# Biliverdin Reductase B Is a Plasma Biomarker for Intraplaque Hemorrhage and a Predictor of Ischemic Stroke in Patients with Symptomatic Carotid Atherosclerosis

**DOI:** 10.3390/biom13060882

**Published:** 2023-05-24

**Authors:** Melody Chemaly, David Marlevi, Maria-Jesus Iglesias, Mariette Lengquist, Malin Kronqvist, Daniel Bos, Dianne H. K. van Dam-Nolen, Anja van der Kolk, Jeroen Hendrikse, Mohamed Kassem, Ljubica Matic, Jacob Odeberg, Margreet R. de Vries, M. Eline Kooi, Ulf Hedin

**Affiliations:** 1Department of Molecular Medicine and Surgery, Karolinska Institutet, 17177 Stockholm, Swedenulf.hedin@ki.se (U.H.); 2Institute for Medical Engineering and Science, Massachusetts Institute of Technology, Cambridge, MA 02142, USA; 3Science for Life Laboratory, Department of Protein Science, School of Engineering Sciences in Chemistry/Biotechnology and Health, KTH Royal Institute of Technology, 11428 Stockholm, Sweden; 4Department of Radiology and Nuclear Medicine, Erasmus MC, University Medical Center Rotterdam, 3015 GD Rotterdam, The Netherlands; 5Department of Epidemiology, Erasmus MC, University Medical Center, 3000 CA Rotterdam, The Netherlands; 6Department of Medical Imaging, Radboud University Medical Center, 6500 HB Nijmegen, The Netherlands; 7Department of Radiology, University Medical Center Utrecht, 3508 GA Utrecht, The Netherlands; 8Department of Radiology and Nuclear Medicine, CARIM School for Cardiovascular Diseases, Maastricht University Medical Center, 6229 ER Maastricht, The Netherlands; 9Department of Hematology, Karolinska University Hospital, Huddinge, 14152 Stockholm, Sweden; 10Department of Clinical Medicine, UiT—The Arctic University of Norway, 9019 Tromsø, Norway; 11Einthoven Laboratory, Department of Surgery, Leiden University Medical Center, 2333 ZA Leiden, The Netherlands; 12Department of Vascular Surgery, Karolinska University Hospital, 17176 Stockholm, Sweden

**Keywords:** intraplaque hemorrhage, biliverdin reductase B, ischemic stroke, vulnerable atherosclerotic plaque, antiangiogenic therapy, magnetic resonance imaging

## Abstract

Background: Intraplaque hemorrhage (IPH) is a hallmark of atherosclerotic plaque instability. Biliverdin reductase B (BLVRB) is enriched in plasma and plaques from patients with symptomatic carotid atherosclerosis and functionally associated with IPH. Objective: We explored the biomarker potential of plasma BLVRB through (1) its correlation with IPH in carotid plaques assessed by magnetic resonance imaging (MRI), and with recurrent ischemic stroke, and (2) its use for monitoring pharmacotherapy targeting IPH in a preclinical setting. Methods: Plasma BLVRB levels were measured in patients with symptomatic carotid atherosclerosis from the PARISK study (*n* = 177, 5 year follow-up) with and without IPH as indicated by MRI. Plasma BLVRB levels were also measured in a mouse vein graft model of IPH at baseline and following antiangiogenic therapy targeting vascular endothelial growth factor receptor 2 (VEGFR-2). Results: Plasma BLVRB levels were significantly higher in patients with IPH (737.32 ± 693.21 vs. 520.94 ± 499.43 mean fluorescent intensity (MFI), *p* = 0.033), but had no association with baseline clinical and biological parameters. Plasma BLVRB levels were also significantly higher in patients who developed recurrent ischemic stroke (1099.34 ± 928.49 vs. 582.07 ± 545.34 MFI, HR = 1.600, CI [1.092–2.344]; *p* = 0.016). Plasma BLVRB levels were significantly reduced following prevention of IPH by anti-VEGFR-2 therapy in mouse vein grafts (1189 ± 258.73 vs. 1752 ± 366.84 MFI; *p* = 0.004). Conclusions: Plasma BLVRB was associated with IPH and increased risk of recurrent ischemic stroke in patients with symptomatic low- to moderate-grade carotid stenosis, indicating the capacity to monitor the efficacy of IPH-preventive pharmacotherapy in an animal model. Together, these results suggest the utility of plasma BLVRB as a biomarker for atherosclerotic plaque instability.

## 1. Introduction

Atheroembolism from atherosclerotic plaques in the carotid bifurcation is an important cause of ischemic strokes [1]. Intraplaque hemorrhage (IPH) has increasingly been recognized as one of the most important plaque morphological features that makes plaques prone to rupture, resulting in subsequent ischemic events [2,3,4]. IPH is believed to be caused by immature and leaky neovessels that invade the plaque from the adventitia, resulting in the release of pro-oxidative hemoglobin (Hb) [5,6], expansion of the lipid-rich necrotic core, and accelerated plaque inflammation [7]. Despite a key role for IPH in the most serious consequences of atherosclerosis, no pharmacotherapy has yet been established for prevention of IPH and plaque instability in humans [8]. Although inhibition of plaque angiogenesis by targeting vascular endothelial growth factor receptor-2 (VEGFR-2) was recently shown to reduce IPH in mice [9], it failed to translate safely into humans [10].

Nevertheless, identifying IPH in clinical practice could aid risk stratification and individualized therapy for patients with carotid stenosis. IPH can be identified by carotid imaging using innate biological properties to guide detection of tissue features [11,12]. In particular, magnetic resonance imaging (MRI) has excellent capacity to differentiate soft-tissue atherosclerotic components [13] and is used as a gold-standard reference for the detection of IPH [14] with clinical potential in stroke risk assessment [15]. Nevertheless, the technology is relatively expensive and has restricted availability, limiting the use of MRI for routine screening in general stroke risk assessment. Instead, plasma biomarkers have been proposed for the identification of patients with IPH and associated unstable lesions [16], guiding the selection of high-risk patients for referral to higher-order diagnostic imaging such as MRI.

Using an inductive approach for the identification of plasma biomarkers causally linked to unstable carotid atherosclerosis based on a biobank of carotid endarterectomies (Biobank of Karolinska Endarterectomies; BiKE, Stockholm, Sweden), we recently discovered biliverdin reductase B (BLVRB) as a candidate biomarker associated with IPH. BLVR is a key enzyme in Hb metabolism, catalyzing the final conversion of biliverdin to bilirubin downstream from heme oxygenase, with the BLVRB isoform typically expressed during fetal development, while BLVRA is predominantly expressed in adult tissues [17]. BLVRB was enriched in plaques and plasma from patients with symptomatic carotid atherosclerosis and was also found to be elevated in a larger cohort of patients with coronary artery disease [18]. Importantly, a causal link among plasma BLVRB levels, IPH, and Hb metabolism was established from correlations of gene and protein expression in carotid plaques, colocalizing BLVRB with HMOX-1 in areas of IPH in plaques from symptomatic patients, as well as in vitro investigations. However, the association of plasma BLVRB with clinically identified IPH, or with stroke risk in patients with carotid atherosclerosis has yet to be established.

The aim of this study was, therefore, to explore the biomarker potential of BLVRB by evaluating the relationship between plasma BLVRB levels and the clinical presence of IPH, as well as the incidence of recurrent ischemic stroke in patients with symptomatic carotid stenosis. Furthermore, to evaluate the capacity of BLVRB to monitor pharmacotherapeutic effects specifically targeting IPH, plasma BLVRB levels were assessed in a preclinical atherosclerotic mouse vein graft model of unstable atherosclerosis subject to antiangiogenic treatment targeting IPH.

## 2. Materials and Methods

### 2.1. Clinical Study

#### 2.1.1. Patient Cohort

The Plaque At RISK (PARISK) study (clinical trials.gov NCT01208025) is a prospective multicenter cohort study that included patients with recent (<3 months) neurological symptoms due to ischemia in the carotid artery territory with a ≥2–3 mm carotid plaque and <70% ipsilateral carotid artery stenosis (based on NASCET criteria) [19] who were not scheduled for carotid endarterectomy or stenting. The study was originally designed to investigate vulnerable plaque features in 300 patients with ipsilateral mild to moderate stenosis using different imaging modalities including MRI, multidetector-row computed tomography angiography (MDCTA), and ultrasonography (US) examination of the carotid arteries and exploration of these data in relation to future ischemic events. Patients were eligible for imaging and blood withdrawal within 3 months after the initial ischemic neurovascular event classified as minor ischemic stroke, transient ischemic attack (TIA), or ocular TIA (amaurosis fugax; AFX). Plasma was isolated after blood withdrawal for further analysis, and 177 blood samples were available from the PARISK cohort for this study. The median number of days between MRI/blood withdrawal and the event was 46 days. The medication status presented in Table 1 reflects the therapy before the qualifying event, which was optimized in all patients after study inclusion. Clinical follow-up to monitor recurrent ischemic events was performed at 3 months follow-up and thereafter yearly for 5 years using structured medical case record forms and/or verified with the general practitioner [20]. The primary endpoint for our study was the occurrence of a recurrent clinical ipsilateral ischemic stroke (defined as ischemic neurovascular symptoms, comprising both minor and major stroke) during 5 years follow-up. The secondary endpoint was the composite of recurrent clinical ipsilateral ischemic stroke, TIA, or AFX during this follow-up. Detailed information from the PARISK study design and primary results can be found elsewhere [19,20,21].

#### 2.1.2. Magnetic Resonance Imaging (MRI)

In the PARISK study, MRI was performed at 3 T using a whole-body scanner with a dedicated carotid coil (four or eight channels), as described previously [19,21]. Six trained observers, all blinded to clinical data and other imaging tests, evaluated the retrieved data and manually segmented the contours of select plaque components (IPH, lipid-rich necrotic core, calcifications, and fibrous tissue) using the VesselMass software (Department Radiology, Leiden University Medical Center, the Netherlands) with a predefined standardized protocol [19]. Specifically, IPH was defined as a hyperintense signal in the bulk of the plaque compared with the adjacent sternocleidomastoid muscle in 3D T_1_-weighted inversion recovery turbo field echo or 3D T_1_-weighted (T_1_w) spoiled gradient echo images and was delineated to calculate the volume [22]. The lipid-rich necrotic core was delineated as an isointense to hyperintense region within the bulk of the plaque on pre-contrast T_1_w MRI that did not enhance on the post-contrast T_1_w MRI. In addition, IPH volume was always considered as part of the lipid-rich necrotic core [23] and was investigated as a categorical variable (presence/absence of IPH). For further details on the above image analysis, please see [19,21].

### 2.2. Animal Study 

#### 2.2.1. Vein Graft Model

The animal study was performed in compliance with Dutch government guidelines and European regulations, with all experiments approved by the animal welfare committee of the Leiden University Medical Center (approval number 12066). Male *ApoE3Leiden* mice, 10–16 weeks old, were fed a hypercholesterolemic diet (ABdiets, Woerden, the Netherlands) from 3 weeks prior to surgery until sacrifice with resulting plasma cholesterol levels between 12 and 24 mmol/L (Roche Diagnostics, kit 1489437, Almere, the Netherlands). Before surgery, mice were anesthetized with midazolam (5 mg/kg, Roche Diagnostics), medetomidine (0.5 mg/kg, Orion, Espoo, Finland), and fentanyl (0.05 mg/kg, Janssen Pharmaceutica, Diegum, Belgium). After surgery, sedation was reversed with atipamezole (2.5 mg/kg, Orion) and fluminasenil (0.5 mg/kg Fresenius Kabi, Schelle, Belgium). Buprenorphine (0.1 mg/kg, MSD Animal Health, Boxmeer, the Netherlands) was given for analgesia. Vein graft surgery was performed by interposition of the inferior vena cava from donor littermate mice in the carotid artery of recipient mice as described before [24]. In brief, the right carotid artery was dissected, cut in the middle, everted around cuffs placed at both ends of the artery, and ligated with 8.0 sutures; the vein graft was thereafter sleeved over the two cuffs and ligated. The atherosclerotic vein graft mouse model has been established for modeling of plaque instability and IPH where lesions display neovascularization with extravasated red blood cells, a necrotic core, and areas of calcification [25]. 

For IPH inhibition, mice were treated with IP injections of rat anti-mouse VEGFR-2 IgG monoclonal blocking antibodies (10 mg·kg^−1^ DC101; Bio X cell, Lebanon, NH, USA) (*n* = 14) or control rat anti-mouse IgG antibodies (10 mg·kg^−1^; *n* = 14; Bio X cell) on days 14, 17, 21, and 25. Two mice in this group were excluded due to vein graft thrombosis [9]. Animals were sacrificed after 28 days and perfusion-fixed for 3 min; the grafts were harvested, fixed in 4% formaldehyde, dehydrated, and paraffin-embedded for histology. The IPH area in the vessel wall was measured and expressed in mm^2^. Plasma samples were collected at T0 before the surgery and randomization and at T28.

#### 2.2.2. Histological and Immunohistochemical Analysis

Paraffin-embedded cross-sections (5 μm) of the vein grafts were processed for Masson’s trichrome and immunohistochemistry staining. In brief, tissue sections were deparaffinized in Histolab clear (Histolab, 14250) and rehydrated in graded ethanol. For antigen retrieval, slides were subjected to high-pressure boiling in DIVA buffer (pH 6.0). After blocking with Rodent block M, (RBM961), primary antibodies BLVRB (HPA041698, Sigma, St Louis, MO, USA) and HMOX1 (HPA000635, Sigma) were diluted in Da Vinci Green solution (Biocare Medical, Pacheco, CA, USA, PD900), applied on slides, and incubated at room temperature for 1 h. For colocalizations, antibodies for cell-specific markers were used: von Willebrand factor (vWF, M0616, DAKO, Glostrup, Denmark), CD68 (ab31630, abcam), and smooth muscle α-actin (SMA, NBP2-33006, Novus, Littleton, CO, USA). A double-stain probe-polymer system containing alkaline phosphatase and horseradish peroxidase was applied, with subsequent detection using Warp Red (Biocare Medical, WR806) and Vina Green (Biocare Medical, BRR807A). Slides were counterstained with Hematoxylin QS (Vector Laboratories, Burlingame, CA, USA), dehydrated, and mounted in Pertex (Histolab, Gothenburg, Sweden). The presence of ferric iron deposits was assessed by Perl’s Prussian blue stain on adjacent serial plaque sections or fixed cells, performed using Perl’s stain kit, (Atom Scientific, Hyde, UK, RRSK 16-100) according to the manufacturers’ instructions, followed by counterstaining with nuclear Fast Red. Images were taken using an Olympus VS200 slide scanner and processed with OlyVIA V3.3 software. 

### 2.3. Analysis of Plasma BLVRB Levels

Plasma BLVRB levels were analyzed from both human and mouse blood samples. The affinity proteomic reagent targeting BLVRB (HPA041937) was obtained from the Human Protein Atlas (HPA) project and was used to detect BLVRB levels as previously described [18]. In brief, HPA041937 antibody was coupled to color-coded magnetic beads (MagPlex^®^, Luminex Corp, Austin, TX, USA). Beads coupled with rabbit IgG from nonimmunized rabbits (Bethyl) and beads without proteins immobilized served as negative controls. EDTA plasma samples were centrifuged and diluted (1:50) in assay buffer (PVXcasein + 10% rabbit IgG) and heated for 30 min at 56 °C, followed by incubation with antibody-coupled and negative control beads overnight. The beads were then washed (PBS/0.05% Tween-20), and bound proteins were crosslinked through the addition of 0.4% paraformaldehyde (PFA). Lastly, streptavidin-conjugated fluorophore (phycoerythrin, Invitrogen, Waltham, MA, USA, SAPE diluted 1:750 in PBS-T) was added as detection reagent. The relative abundance of proteins was measured in FlexMap3D instrument (Luminex Corp.), and the resulting median fluorescence intensity (MFI) per bead identity was used for data processing. Samples were randomized and run in duplicate in a 96-well plate, and measurements were averaged for each sample. Samples with a high reactivity with rabbit IgG beads were excluded from the analysis. BLVRB levels are expressed as the mean fluorescence intensity using arbitrary units. Optimization of this method was performed in our previous publication in the supplementary files (Supplementary Figure IV) [18].

### 2.4. Statistical Analysis

Statistical analysis was performed using IBM^®^ SPSS software version 28.0.0.0 (SPSS Inc., Chicago, IL, USA). Patients with baseline blood samples taken no more than 3 months following their neurological symptoms were included. Patient plasma samples with cross-reactivity with the rabbit IgG antibody were also excluded from the analysis (Figure 1). BLVRB plasma level values were standardized (z-score) following kurtosis and skewness tests to allow the application of linear statistics and calculate OR, HR, and CI. Pearson’s chi-square test was used to compare proportions in two groups, and univariable analysis (Student’s *t*-test) was conducted to compare BLVRB levels between patient groups. Logistic regression analysis was applied to evaluate the association of BLVRB levels with the presence of IPH adjusted for age and sex. Pearson’s correlation analysis was performed to explore associations between BLVRB levels and continuous clinical parameters measured in this study. Cox regression was used to evaluate the association between BLVRB levels and the occurrence of a recurrent clinical ipsilateral ischemic stroke during 5 years follow-up (primary endpoint), as well as the occurrence of recurrent clinical ipsilateral ischemic stroke, TIA, or AFX during 5 years follow-up (secondary endpoint). Kaplan–Meier curves and life tables were generated to describe survival probabilities for the primary and secondary endpoints. In the mouse study, a paired t-test was used to compare plasma BLVRB levels between baseline and the end of the study separately within the VEGFR-2- and IgG-treated groups. Statistical significance was defined at *p* < 0.05 (two-tailed).

## 3. Results

### 3.1. Patient Demographics and IPH

A total of 177 patients with blood sample symptomatic, low- to moderate-grade, carotid stenosis from the PARISK cohort were included for evaluation. Of these, 16 patients were excluded due to the unavailability of the blood sample or MRI at baseline (Figure 1). In the remaining 161 patients, IPH was detected in MRI in 65 (40.4%) subjects (Table 1). Among the patients with IPH, a significant proportion were male compared to females (89.2% vs. 10.8%; *p* < 0.001). There was a nonsignificant trend for more IPH in patients admitted with stroke compared to patients with TIA or patients with AFX (55.4% vs. 32.3% vs. 12.3%; *p* = 0.071) (Table 1). However, in patients with hemispheric symptoms only, those admitted with stroke had a significantly higher percentage of IPH compared to patients admitted with TIA (63.2% vs. 36.8%; *p* = 0.022), s well as in patients with stroke compared to those with TIA or AFX combined (55.4% vs. 44.6%; *p* = 0.035) (Table 1). 

Patients with IPH were also more often former smokers (67.7% vs. 46.3%, *p* = 0.027) and had a significantly higher degree of stenosis estimated by the NASCET criteria (22.6 ± 17.2% vs. 13.6 ± 16.3%, *p* = 0.002) compared to patients with no IPH. Of note, LDL-C levels were lower in patients with IPH compared to patients without IPH (2.9 ± 1.0 vs. 3.4 ± 1.00 mmol/L; *p* = 0.033) (Table 1). 

### 3.2. Plasma BLVRB Levels Are Significantly Higher in Patients with Carotid Plaque IPH

From the patients included in the above statistical analysis (*n* = 161), 20 blood samples showing high reactivity with the detection antibody (rabbit IgG) were excluded from the plasma analysis (Figure 1). In the remaining samples (*n* = 141), plasma BLVRB levels were significantly higher in patients with IPH compared to those without IPH (737.3 ± 693.2 vs. 520.9 ± 499.4 mean fluorescence intensity (MFI), *p* = 0.033) (Figure 2A, Table 2). In contrast, plasma BLVRB levels were not significantly different between males and females, nor were they affected by smoking status, the presence of comorbidities, or medication at the time of admission (Table 2). In addition, plasma BLVRB levels did not significantly correlate with age, BMI, C-reactive protein levels, LDL-C levels, or stenosis degree. In a model including age and sex, the association between plasma BLVRB levels and IPH was nonsignificant (OR = 1.418; CI [0.990–2.032], *p* = 0.057) due to the higher presence of IPH in men vs. women (89.2% vs. 61.5%; *p* < 0.001). However, plasma BLVRB levels were not significantly different between males and females (*p* = 0.330, Table 2).

Of note, plasma BLRVB levels did not correlate with IPH volume in mm^3^ (r = 0.055; *p* = 0.515) or other parameters assessed by MRI including the lipid-rich necrotic core size (r = 0.079, *p* = 0.354), volume of fibrous tissue (r = 0.009, *p* = 0.912), or calcification volume (r = 0.132, *p* = 0.120).

We found no association between the severity of symptoms upon admission (stroke, TIA, and AFX) and plasma BLVRB levels (*p* = 0.223) (Table 2). Likewise, although plasma BLVRB levels tended to be higher in patients admitted with stroke or TIA (*n* = 126) compared to patients admitted with AFX (*n* = 15), this did not infer statistical significance (628.4 ± 616.2 vs. 397.2 ± 114.8 MFI, *p* = 0.151). However, for patients admitted with stroke or TIA upon admission only (excluding those admitted with AFX), patients with IPH had significantly higher levels of plasma BLVRB levels compared to those without IPH (711.7 ± 664.7 vs. 557.0 ± 537.6 MFI, *p* = 0.033).

### 3.3. Plasma BLVRB Levels Associate with Recurrent Ischemic Stroke

During the 5 year follow-up, 7.8% (*n* = 11) of the patients developed a recurrent ischemic stroke. These patients had significantly higher levels of plasma BLVRB on admission compared to those without recurrent ischemic stroke (1099.3 ± 928.5 vs. 582.1 ± 545.3 MFI, HR = 1.600, CI [1.092–2.344]; *p* = 0.016). This association remained statistically significant following adjustment for age and sex (HR = 1.607, CI [1.072–2.409]; *p* = 0.022) (Figure 2B,C). Of note, for the primary endpoint, no females experienced recurrent ischemic stroke. Patients who developed the composite endpoint of ischemic stroke, TIA, or AFX (17.7% of the total number of patients, out of which 62.31% had IPH on admission) also had higher plasma BLVRB levels upon admission compared to those who did not (852.9 ± 778.7 vs. 573.8 ± 540.6 MFI, HR = 1.348, CI [1.010–1.799]; *p* = 0.042); however, this association did not remain statistically significant once adjusted for age and sex (*p* = 0.075) (Figure 2D,E).

Stratification of the secondary endpoint to associate plasma BLVRB levels with recurrent ischemic stroke, TIA, or AFX separately showed that plasma BLVRB levels were significantly predictive of recurrent ischemic stroke (as mentioned above, *p* = 0.016) but not recurrent TIA (*p* = 0.213) or recurrent AFX (*p* = 0.225). However, plasma BLVRB levels significantly associated with the combined endpoint of recurrent ischemic stroke and TIA after excluding AFX (HR = 1.469, CI [1.094–1.974]; *p* = 0.011, adjusted for age and gender; *p* = 0.038). 

Next, we investigated if the severity of symptoms upon admission was associated with plasma BLVRB levels and recurrent events. In patients admitted with a stroke or TIA (*n* = 126), BLVRB levels significantly predicted the secondary composite outcome (recurrent ischemic stroke, TIA or AFX) (HR = 1.409, CI [1.048–1.895]; *p* = 0.023), but not in those admitted with AFX (*n* = 15), (*p* = 0.503). Of note, no patients admitted with AFX experienced recurrent ischemic stroke upon follow-up.

### 3.4. Plasma BLVRB Levels Reflect Inhibition of IPH in Mice

In the mouse vein graft model, plasma BLVRB levels were significantly lower in the VEGFR-2 inhibitor group compared to baseline (1189 ± 258 vs. 1752 ± 366 MFI; *p* = 0.004), whereas no significant difference was seen in the IgG-treated group (1271 ± 163 vs. 1558 ± 312; *p* = 0.116) compared to baseline (Figure 3A). Furthermore, qualitative assessment of Masson trichrome-stained sections indicated reduced areas of IPH and higher collagen content in the VEGFR-2 inhibitor group (Figure 3B). In addition, immunohistochemical staining of vein grafts from control mice showed colocalization of BLVRB with CD68 positive macrophages in regions of low smooth muscle alpha actin expression (SMA) and heme oxygenase 1 (HMOX1) (Figure 4), confirming the previously shown association of BLVRB to Hb metabolism [18]. Additionally, Perl’s and vWF immunohistochemical staining in vein grafts from control mice showed IPH and neovessels in control mice (Appendix A), whereas IPH area was reduced in the VEGFR-2-treated group (1.7 ± 0.8 vs. 1.1 ± 0.3; *p* = 0.051) (Appendix A). Of note, there was no significant difference in BLVRB plasma levels at baseline between VEGFR-2 inhibitor and IgG control groups (*p* = 0.272), where plasma samples were collected before the surgery and before randomization.

## 4. Discussion

In this study, we evaluated the potential of plasma BLVRB as a biomarker for IPH, plaque instability, and recurrent ischemic cerebral events across multiple platforms. First, *in a clinical setting*, we demonstrated that plasma BLVRB levels were associated with the presence of IPH in patients with symptomatic, low- to moderate-grade carotid stenosis and with recurrent ischemic stroke during a 5 year follow-up. Second, *in a preclinical setting*, we showed reduced plasma BLVRB levels following VEGFR-2 inhibition of IPH, highlighting the ability to monitor response to pharmacotherapy targeting plaque neovessel formation and IPH. Together, these results strengthen the potential of plasma BLVRB as a future biomarker for plaque instability.

### 4.1. Plasma BLVRB Levels Are Associated with IPH in Patients with Symptomatic Carotid Atherosclerosis

First, we found an association between plasma BLVRB levels and the presence of IPH, as assessed by MRI in patients with low- to moderate-grade symptomatic carotid stenosis. This finding suggests that plasma BLVRB measurement can be used for identification of high-risk patients, as the detection of IPH by MRI was previously shown to predict stroke in patients with carotid stenosis [15,26], as well as recently demonstrated in our PARISK cohort [22]. In this cohort, we also found that the presence of IPH on MRI was associated with more severe, hemispheric symptoms and a higher degree of stenosis at baseline, which is in line with the predictive value established for these parameters [27] and in support of the validity of the study cohort. In addition, there was also a trend toward higher plasma BLVRB levels in patients admitted with more severe symptoms, which possibly did not reach significance due to a low sample number. Of note, the observation that patients with low LDL-C presented with IPH suggests residual risk despite lipid-lowering therapy [28], which would further motivate the need for improved risk estimation in these patients.

### 4.2. Plasma BLVRB Levels Predict Recurrent Ischemic Stroke in Patients with Symptomatic Carotid Atherosclerosis

Previously, we reported elevated plasma BLVRB levels in patients with symptomatic vs. asymptomatic carotid stenosis, as well as an association with increasing coronary atherosclerosis as stratified by coronary artery calcification scoring [19]. Here, we evaluated the association between BLVRB levels and future risk of stroke. The predictive value of BLVRB was strengthened by demonstrating that high BLVRB levels upon admission were associated with recurrent ischemic stroke during a 5 year follow up. Of note, 36.4% of the patients who developed a recurrent ischemic stroke did not have an IPH detected by MRI (data not shown), suggesting a higher precision of BLVRB plasma levels in predicting recurrent ischemic strokes compared with IPH detected by MRI. Furthermore, BLVRB plasma levels were not confounded by other clinical parameters such as comorbidities, medication status or biochemical measurements, which together strengthens the specificity of BLVRB for plasma level IPH detection. Plasma BLVRB levels may, thus, provide added value to current stroke risk predictors and aid in clinical decision making [11]. With higher-order MRI being a comparably resource-demanding utility [29], plasma biomarkers such as BLVRB may represent a low-cost alternative and a viable high-throughput screening method for IPH detection, offering a pre-stratification step of patients for further clinical investigation. The long-term outcome data also provided further support for the validity of the study cohort, as plasma BLVRB levels predicted recurrent ischemic events in patients admitted with hemispheric symptoms rather than in those admitted with AFX, previously shown to be associated with low risk for recurrent events [30]. 

### 4.3. Plasma BLVRB Measurement Can Be Used to Monitor Response to Antiangiogenic Therapy

In a dedicated preclinical setup, we showed how plasma BLVRB levels can be successfully used to monitor pharmacotherapy geared toward modulation of IPH through antiangiogenic therapy, a strategy previously shown to inhibit atherosclerotic plaque neovessel formation and IPH in animal models [31,32]. Here, we tested the effect of VEGFR-2 inhibition on plasma BLVRB levels in a mouse model of IPH, in which VEGFR-2 inhibition was previously shown to attenuate intralesional angiogenesis and IPH [9]. Vein grafts in hypercholesterolemic ApoE3Leiden mice display features of plaque instability and rupture, including IPH and angiogenesis, which are rarely seen in other murine atherosclerotic models [25]. Here, VEGFR-2 inhibition was associated with reduced plasma BLVRB levels, most likely as a consequence of decreased IPH within the vein grafts. In a similar manner to human atherosclerosis [18], BLVRB and its upstream enzyme in Hb catabolism, HMOX1, were also expressed in regions of CD68-positive macrophages in atherosclerotic mouse vein grafts. These results provide further support for a close association between plasma BLVRB levels and IPH, as well as suggest that plasma BLVRB levels can be used as a biomarker to monitor response to antiangiogenic therapy in the application of such strategies for the development of plaque-stabilizing pharmacotherapy. 

### 4.4. Limitations

A few limitations should be pointed out. First, a comparably small cohort size of symptomatic patients was utilized, along with a low number of recurrent ischemic events recorded, warranting replication of our findings in larger cohorts. Additionally, as seen in many other CVD trials [33], our cohort suffered from an under-representation of women. This, along with the fact that women seem less likely to present with IPH compared to men, also requires further investigation. The association between plasma BLVRB levels and IPH was further marginally confounded by sex and gender, possibly due to a small sample size and the significantly higher percentage of IPH in men than women, as previously reported [34]. Of note, the study cohort consisted of patients with low- to moderate-grade carotid stenosis and may not be representative for patients with more significant stenosis. Nevertheless, as patients with ischemic cerebral symptoms and a carotid stenosis below 50% (NASCET) are not recommended for intervention by most guidelines [35] and rather categorized as suffering from ’embolic stroke of unknown source’ [36], these patients would likely benefit the most from a predictive biomarker, such as plasma BLVRB, to guide clinical decision making. Lastly, it should be noted that the assessment of changes in plasma BLVRB levels following VEGFR-2 was performed exclusively in a preclinical mouse model. Whilst IPH treatment in animal models has previously failed to translate into humans [10], our study still indicates how plasma BLVRB levels could be used as an effective indicator of drug response during the development of novel antiangiogenic therapies.

## 5. Conclusions

This study strengthens the potential value of plasma BLVRB as a biomarker for unstable carotid atherosclerosis in multiple ways: first, by demonstrating an association between plasma BLVRB and IPH in a clinical cohort of patients with symptomatic carotid atherosclerosis; second, by indicating an independent association with the incidence of recurrent ischemic stroke in the same cohort; third, by showing reduced plasma BLVRB levels following inhibition of plaque neovascularization and IPH development in a preclinical mouse model. Together, these results strengthen the potential clinical utility of plasma BLVRB measurement in ischemic stroke risk prediction of patients with carotid atherosclerosis, as well as in monitoring of drug response during the development of pharmacotherapy for plaque stabilization and prevention of IPH. 

## 6. Clinical Perspectives

IPH is a vulnerable atherosclerotic plaque feature and a major predictor of cardiovascular events. IPH detection is possible by MRI but remains expensive. Plasma BLVRB measurement could offer added value for IPH detection and ischemic stroke risk prediction of patients with carotid atherosclerosis. Moreover, during the development of antiangiogenic pharmacotherapies for prevention of IPH and plaque stabilization, plasma BLVRB could potentially be used to monitor drug response.

## Figures and Tables

**Figure 1 biomolecules-13-00882-f001:**
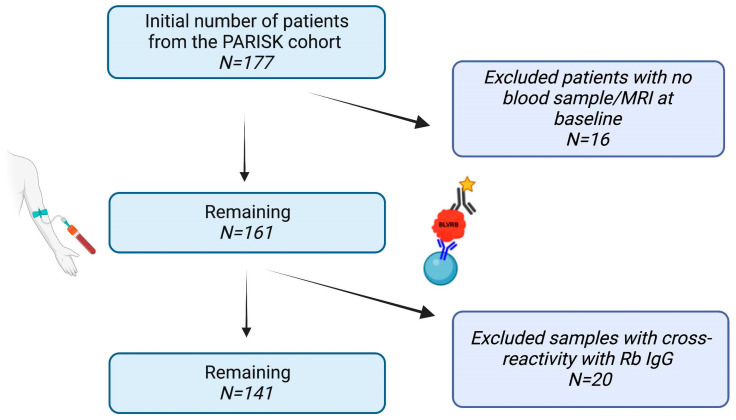
Workflow for sample analysis. BLVRB: biliverdin reductase B, MRI: magnetic resonance imaging.

**Figure 2 biomolecules-13-00882-f002:**
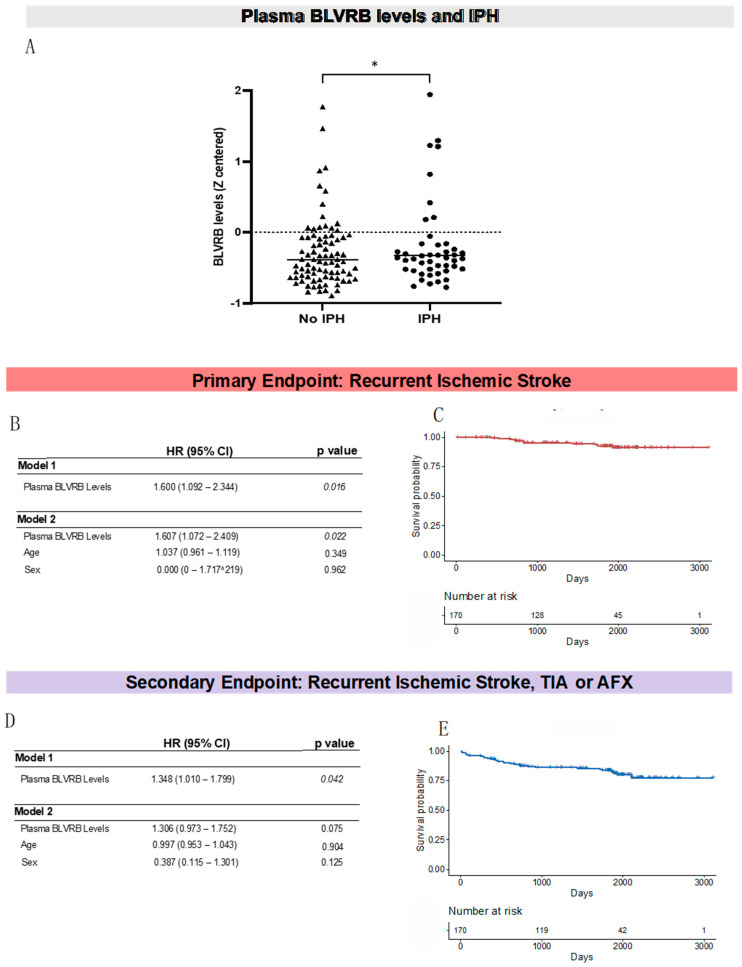
Associations of BLVRB levels with IPH and recurrent ischemic stroke. (**A**) BLVRB plasma levels in symptomatic patients with low- to moderate-grade carotid stenosis and IPH compared to those with no IPH as assessed by carotid MRI. (**B**) Cox proportional hazard regression analysis of plasma BLVRB levels and the primary outcome of recurrent ischemic stroke. (**C**) Kaplan–Meier survival curve and life table for the primary outcome of recurrent ischemic stroke. (**D**) Cox proportional hazard regression analysis of plasma BLVRB levels and the secondary outcome of recurrent ischemic stroke, TIA, or AFX. (**E**) Kaplan–Meier survival curve and life table for the secondary outcome of recurrent ischemic stroke, TIA, or AFX. Plasma BLVRB levels expressed as the median fluorescence intensity (MFI) with mean ± SD in arbitrary units. AFX: amaurosis fugax, BLVRB: biliverdin reductase B, TIA: transient ischemic attack. * Refers to *p* < 0.05.

**Figure 3 biomolecules-13-00882-f003:**
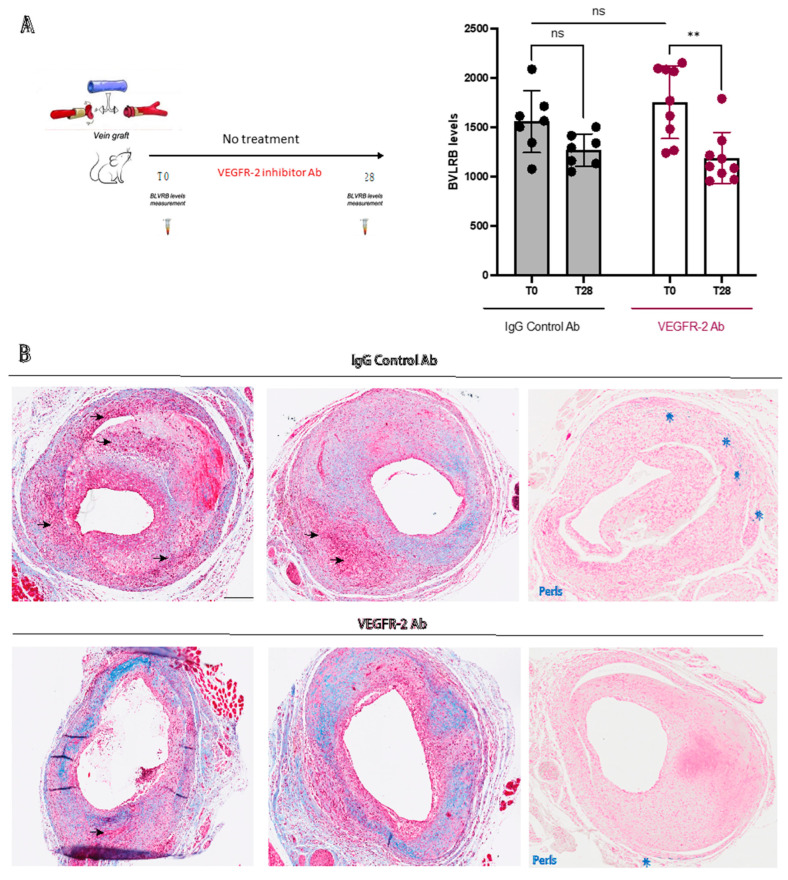
Evaluation of plasma BLVRB levels and vein graft histology following VEGFR-2 inhibition. (**A**) Plasma BLVRB levels in atherosclerotic vein graft mouse model with or without treatment with VEGFR-2 inhibitor Ab. (**B**) Masson trichrome and Perl’s staining of mouse vein grafts from IgG control Ab mice compared to mice treated with VEGFR-2 Ab at the T28 timepoint. Note the reduction in IPH (arrowheads) and increased staining for collagen (blue) in the treatment group, as well as the reduction in Perl’s staining (blue stars) in the treatment group compared to the control. BLVRB: biliverdin reductase B, IPH: intraplaque hemorrhage, VEGFR-2: vascular endothelial growth factor receptor 2. ** Refers to *p* < 0.01 and ns refers to non-significant. Scale bar indicates 200 μm.

**Figure 4 biomolecules-13-00882-f004:**
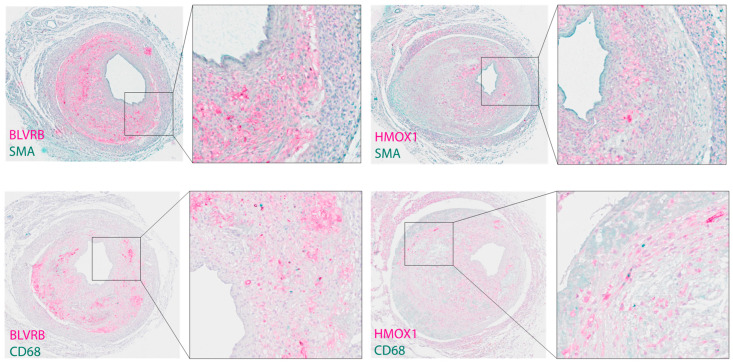
Immunohistochemical double staining for BLVRB and HMOX1 with SMA and CD68 in mouse vein grafts. Note the colocalization of BLVRB and HMOX1 and concentration in regions of low SMA and high CD68 expression. BLVRB: biliverdin reductase B, HMOX1: heme oxygenase 1, SMA: smooth muscle alpha actin.

**Table 1 biomolecules-13-00882-t001:** Patient demographics with detected intraplaque hemorrhage (IPH) by MRI.

Baseline Characteristics	No IPH (*n* = 96)	IPH (*n* = 65)	*p*-Value *
	(%)	(%)	
Male	61.5	89.2	<0.001
Baseline event between groups:			
Ischemic stroke	38.5	55.4	0.071
TIA	50.0	32.3	
AFX	11.5	12.3	
Hemispheric symptoms only:			
Ischemic stroke	43.5	63.2	0.022
TIA	56.5	36.8	
Major compared to minor events:			
Ischemic stroke	38.5	55.4	0.035
TIA or AFX	61.5	44.6	
Smoking			
Never	28.4	18.5	0.027
Former	46.3	67.7	
Current	25.3	13.8	
Hypertension	63.3	73.8	0.156
Diabetes mellitus	21.9	23.1	0.857
Hypercholesterolemia	59.8	69.2	0.225
Antihypertensive therapy	58.3	64.6	0.423
Antidiabetic drug use	18.8	18.5	0.963
Statin drug use	49	50	0.897
Antiplatelet drug use	36.5	51.6	0.058
Oral ACO drug use	4.2	0	0.096
Stenosis degree (NASCET)	13.6 ± 16.3	22.6 ± 17.2	0.002
	Mean ± SD	Mean ± SD	*p* value
Age (years)	68.1 ± 9.5	70.7 ± 7.7	0.068
BMI (kg/m^2^)	26.9 ± 4.3	26.4 ± 4.3	0.464
CRP levels (mg/L)	3.6 ± 5.3	2.7 ± 6.9	0.397
LDL-C (mmol/L)	3.4 ± 1.0	2.9 ± 1.0	0.033

* Pearson’s asymptotic sig. two-sided. The medication status presented in Table 1 reflects the therapy before the symptomatic event, whereas medical therapy was optimized in all patients after study inclusion. ACO: anticoagulant, AFX: amaurosis fugax, BMI: body mass index, CRP: C-reactive protein, IPH: intraplaque hemorrhage, LDL-C: low-density lipoprotein cholesterol, MRI: magnetic resonance imaging, NASCET: The North American Symptomatic Carotid Endarterectomy Trial, TIA: transient ischemic attack.

**Table 2 biomolecules-13-00882-t002:** Association between BLVRB levels and clinical parameters.

Parameters		*N*	Mean	Std. Deviation	Q1–Q3	* *p*-Value
Sex						
	Male	103	633.2	605.1	287.0–644.8	0.330
	Female	38	524.0	540.2	227.7–567.1	
Clinical admission						
	Ischemic stroke	66	579.8	527.4	273.2–661.5	0.223
	TIA	60	681.9	701.8	263.5–632.4	
	AFX	15	397.2	114.8	258.3–505.9	
Smoking						
	Never	32	756.3	674.3	289.7–1135.9	0.164
	Former	78	525.7	508.2	227.0–567.5	
	Current	30	641.7	675.5	271.7–613.0	
Hypertension						
	No	43	626.4	692.5	262.9–574.4	0.761
	Yes	97	593.4	543.0	268.1–628.1	
Diabetes mellitus						
	No	107	634.7	626.5	285.5–574.4	0.271
	Yes	34	506.7	442.3	221.1–543.5	
Hypercholesterolemia						
	No	52	672.7	749.2	228.2–707.2	0.360
	Yes	86	577.3	472.5	310.4–593.7	
Antihypertensive therapy						
	No	53	731.1	735.0	268.5–786.2	0.120
	Yes	88	527.2	467.5	226.6–564.8	
Antidiabetic drug use						
	No	112	636.2	614.8	285.9–583.4	0.220
	Yes	29	478.7	461.1	211.1–629.1	
Statin drug use						
	No	69	670.0	675.8	253.5–684.2	0.270
	Yes	71	543.9	489.4	282.6–584.1	
Antiplatelet drug use						
	No	81	587.0	580.0	233.7–582.6	0.656
	Yes	59	632.2	607.3	299.4–636.0	
Oral ACO drug use						
	No	137	604.7	596.3	264.0–581.1	0.914
	Yes	4	572.2	199.8	361.9–722.6	
Stenosis Degree median (NASCET)						
	Stenosis < 30%	87	605.4	618.9	248.1–581.2	0.907
	Stenosis > 30%	37	619.4	594.1	269.5–579.3	
IPH						
	Absent	87	520.9	499.4	237.8–581.1	0.033
	Present (>0)	54	737.3	693.2	300.9–910.5	

Plasma BLVRB levels expressed as the median fluorescence intensity (MFI) with mean ± SD in arbitrary units. The threshold 30% stenosis degree (NASCET) was used, which corresponds to 65% according to ECST2. * Pearson’s asymptotic sig. two-sided. ACO: anticoagulant, AFX: amaurosis fugax, BLVRB: biliverdin reductase B, NASCET: The North American Symptomatic Carotid Endarterectomy Trial, TIA: transient ischemic attack.

## Data Availability

All the original data from this work can be provided upon reasonable request by contacting the authors.

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
