# Peer review of "Biliverdin Reductase B Is a Plasma Biomarker for Intraplaque Hemorrhage and a Predictor of Ischemic Stroke in Patients with Symptomatic Carotid Atherosclerosis"

_biomolecules, 2023, doi:10.3390/biom13060882_

Round 1

Reviewer 1 Report

I congrulate the authors to their manuscript. It covers an important topic, is presented clearly, and well written.

I have only some minor issues:

- I recommend to shorten the introduction with more focus on Biliverdinreductase B.

- In the Results, I would round the stenotic degree of internal carotid artery stenosis to full digits. Same with the levels of BLVRB which should be rounded to full digits or even to full multiples of ten. Use of decimals prtends a pseudo-accuracy.

- It is not clear to me, why in the preclinical approach BLVRB levels at baseline differ between both groups by >10%. BLVRB levels at day 28 are similar so the significance is driven by the differing baseline values, which however, should be identical. Even though it may not be statistical significant, small differences at baseline and at day 28 lead to a significant difference at all. The authors should comment on that. To my mind the conclusion, that treatment with a VEGFR-2 inhibitor reduces BLVRB levels is not clearl supported by the data and should be attenuated.

Author Response

Reviewer 1 

I congrulate the authors to their manuscript. It covers an important topic, is presented clearly, and well written. 

I have only some minor issues: 

- I recommend to shorten the introduction with more focus on Biliverdinreductase B. 

The authors thank the Reviewer for this suggestion. The introduction has now been revised as suggested by the Reviewer. In brief, the introduction has been condensed in the first page, reference 8 removed, and the section on BLVRB expanded.   

- In the Results, I would round the stenotic degree of internal carotid artery stenosis to full digits. Same with the levels of BLVRB which should be rounded to full digits or even to full multiples of ten. Use of decimals prtends a pseudo-accuracy. 

We appreciate this comment by the Reviewer and as suggested, all numbers in the study are now rounded to 1 decimal. Changes are highlighted in yellow.  

- It is not clear to me, why in the preclinical approach BLVRB levels at baseline differ between both groups by >10%. BLVRB levels at day 28 are similar so the significance is driven by the differing baseline values, which however, should be identical. Even though it may not be statistical significant, small differences at baseline and at day 28 lead to a significant difference at all. The authors should comment on that. To my mind the conclusion, that treatment with a VEGFR-2 inhibitor reduces BLVRB levels is not clearl supported by the data and should be attenuated. 

We acknowledge the concerns raised by the Reviewer and would like to comment as follows: 

We have compared the differences at baseline levels between the treated and untreated group and found no significant difference (p=0.272). In the revised version of the manuscript, this has now been updated in Figure 3. In addition, baseline levels of BLVRB were measured in both groups before surgery and before randomization into either treatment or no treatment group. The statistical test which was performed in this comparison was a paired T test which compares two timepoints within the same group and relies on the delta change compared to baseline within the same group and does not compare the levels between the two groups (treated vs untreated). The paired T test was significant only in the treated group and not in the untreated group, which means that the delta change in BLVRB levels compared to the respective baseline was significant in the treated group only. We have now added the following explanation in the manuscript body: 

Methods: 

Plasma samples were collected at T0 before the surgery and randomisation, and at T28 

Statistical analysis: 

In the mouse study, a paired t-test was used to compare plasma BLVRB levels between baseline and at the end of the study within each of the VEGFR-2 and IgG treated groups separately. 

Results: 

Of note, there was no significant difference in BLVRB plasma levels at baseline between VEGFR-2 inhibitor and IgG control groups (p=0.272) where plasma samples were collected before the surgery and before randomisation. 

Reviewer 2 Report

The authors aimed to explore the association between plasma BLVRB and IPH in carotid plaques and to monitor pharmacotherapy targeting IPH in a preclinical setting. Since BLVRB was discovered by the authors through a multi-omics approach, they performed a case-control study to examine the first purpose and an animal study to investigate the second purpose. This study has enormous clinical implications and could provide evidence that plasma BLVRB can be a biomarker to replace MRI for the detection of IPH, a vulnerable atherosclerotic plaque feature. Some questions require attention.

1.     For the BLVRB, the samples were run in duplicates. The results of quality assurance and control results need to illustrate in the paragraph.

2.     BLVRB values were standardized to allow the application of linear statistics and regression models. Thus BLVRB values might not follow the normal distribution; therefore, it should be examined using non-parametric methods.

3.     Results: the data description written in the paragraphs should be presented in tables and figures. For example, in patients with hemispheric symptoms only, those admitted with stroke had significantly higher percentage of IPH compared to patients admitted with TIA (63.2% vs 36.8%; p=0.022)… in section 3.1; BLVRB were not significantly correlated with either age, BMI, CRP, LDL, … in section 3.2, etc.

4.     LDL-C levels were lower in patients with IPH ….. (p=0.003). The p-value shown in this paragraph do not agree with that shown in table 1.

Author Response

Reviewer 2: 
Comments and Suggestions for Authors 

The authors aimed to explore the association between plasma BLVRB and IPH in carotid plaques and to monitor pharmacotherapy targeting IPH in a preclinical setting. Since BLVRB was discovered by the authors through a multi-omics approach, they performed a case-control study to examine the first purpose and an animal study to investigate the second purpose. This study has enormous clinical implications and could provide evidence that plasma BLVRB can be a biomarker to replace MRI for the detection of IPH, a vulnerable atherosclerotic plaque feature. Some questions require attention. 

1.     For the BLVRB, the samples were run in duplicates. The results of quality assurance and control results need to illustrate in the paragraph. 

The authors thank the reviewer for highlighting this. The quality control of BLVRB measurements were performed in our previous publications and highlighted in the supplementary material. This has now been added into the methods section and highlighted in yellow.  

https://www.ncbi.nlm.nih.gov/pmc/articles/PMC6115646/#appsec1 

2.     BLVRB values were standardized to allow the application of linear statistics and regression models. Thus BLVRB values might not follow the normal distribution; therefore, it should be examined using non-parametric methods. 

We thank the Reviewer for this suggestion. We would like to argue that non-parametric methods are less robust as parametric methods especially when it comes to multivariate regression models and survival analysis. Therefore, we chose to standardize BLVRB measurements in order to apply linear statistics which improves predictive capacity when it comes to clinical biostatistics. 

3.     Results: the data description written in the paragraphs should be presented in tables and figures. For example, in patients with hemispheric symptoms only, those admitted with stroke had significantly higher percentage of IPH compared to patients admitted with TIA (63.2% vs 36.8%; p=0.022)… in section 3.1; BLVRB were not significantly correlated with either age, BMI, CRP, LDL, … in section 3.2, etc. 

We thank the Reviewer for this very relevant suggestion. The authors have now integrated this data into table 1 and highlighted. The correlations were more difficult to integrate into Table 2 and were consequently omitted but can be included as graphs in the supplementary material if preferred. 

4.     LDL-C levels were lower in patients with IPH ….. (p=0.003). The p-value shown in this paragraph do not agree with that shown in table 1. 

We apologize for this mistake, which has now been corrected to p=0.033 in the text.  

Reviewer 3 Report

I am delighted to review the study “Biliverdin reductase B is a plasma biomarker for intraplaque hemorrhage and a predictor of ischemic stroke in patients with symptomatic carotid atherosclerosis”. This study aimed to explore the biomarker potential of plasma BLVRB by its correlation with IPH in carotid plaques assessed by magnetic resonance imaging (MRI), and with recurrent ischemic stroke; and its capacity to monitor the efficacy of IPH-preventive pharmacotherapy. Limitations of the study are listed in the corresponding section.

Concern regarding the registration as “Clinical trials.gov NCTO 1208025”: No relevant study found at the link provided.

Clarify please how 300 patients mentioned in the Method section are corresponded with 177 in the Result section. These patients are not reflected in Fig 1 representing the Workflow for sample analysis.

Author Response

Reviewer 3: 

Comments and Suggestions for Authors 

I am delighted to review the study “Biliverdin reductase B is a plasma biomarker for intraplaque hemorrhage and a predictor of ischemic stroke in patients with symptomatic carotid atherosclerosis”. This study aimed to explore the biomarker potential of plasma BLVRB by its correlation with IPH in carotid plaques assessed by magnetic resonance imaging (MRI), and with recurrent ischemic stroke; and its capacity to monitor the efficacy of IPH-preventive pharmacotherapy. Limitations of the study are listed in the corresponding section. 

Concern regarding the registration as “Clinical trials.gov NCTO 1208025”: No relevant study found at the link provided. 

The link for the clinical trial is NCT01208025 which has now been amended in the manuscript.  

Clarify please how 300 patients mentioned in the Method section are corresponded with 177 in the Result section. These patients are not reflected in Fig 1 representing the Workflow for sample analysis. 

We thank the Reviewer for noticing this. Although the PARISK study included 300 patients, blood samples were available from 177 patients and were available for BLVRB measurement for current study. This has now been added to the methods section and highlighted in yellow.  
